

# A hybrid extraction model for semantic knowledge discovery of water conservancy big data

Yanna Feng[1], Feng Zhang[2], Yongheng Zhang[2], Jiangang Dong[2] and PengJu Wang[3]

[1] School of Management, Yulin University, Yulin, Shannxi, China
[2] School of Information Engineering, Yulin University, Yulin, Shannxi, China
[3] Office of the Organizational Committee, Yan'an Municipal Committee, Yanan, Shannxi, China

## ABSTRACT

To address the growing demand for efficient public opinion analysis in water conservancy and related domains, as well as the inefficiencies and limited scalability of existing automated web data extraction algorithms for multi-source datasets, this research integrates advanced technologies including big data analytics, natural language processing, and deep learning. A novel, transferable web information extraction model based on deep learning (WIEM-DL) is proposed, leveraging knowledge graphs, machine learning, and ontology-based methods. This model is designed to adapt to varying website structures, enabling effective cross-website information extraction. By refining water conservancy-related online public opinion content and extracting key feature information from critical sentences, the WIEM-DL model excels in locating main content while filtering out noise. This approach not only reduces processing time but also significantly improves extraction accuracy and efficiency. Furthermore, the model establishes methods for micro-level public opinion information extraction and feature representation, creating a fusion space for data-level integration. This serves as a robust foundation for multi-granularity semantic knowledge integration in public opinion big data. Experimental results demonstrate that the WIEM-DL model substantially outperforms traditional information extraction methods, setting a new benchmark for extraction performance.

## INTRODUCTION

The explosive growth of social-media discourse on water-conservancy topics has created vast, heterogeneous text streams challenging to process with traditional ontology-driven methods. Such methods often rely on hand-crafted rules and features, which are labor-intensive and brittle when page layouts or domain vocabularies change—for example, a single HTML structure update can break document object model (DOM)-tree extraction scripts and require extensive re-engineering (*Chowdhury, Abdullah & Albashrawi, 2024*). They also demand large annotated datasets to maintain precision, making them cost-prohibitive for emerging or niche subdomains like reservoir safety or

Corresponding author
Feng Zhang,
zhangfeng@yulinu.edu.cn

urban flood management. Moreover, these approaches typically target coarse page segments rather than fine-grained semantic units, resulting in low precision for tasks such as sentiment-aware concept extraction. For instance, manually defined templates may identify "reservoir" and "pollution" as entities but fail to capture the sentiment relation "public concern about reservoir pollution," which is crucial for decision support. Consequently, existing systems achieve high accuracy only within narrowly scoped domains and struggle to generalize across new sites or topic shifts.

Information extraction (IE) converts unstructured text into structured triples—⟨Head Entity, Relation, Tail Entity⟩—to facilitate downstream tasks like knowledge-graph construction and event detection. IE can be domain-specific, relying on expert rules (*e.g.*, extracting "dam failure risk" relations from technical reports), or open-domain, extracting general facts without pre-defined schemas. Domain-specific IE often yields high precision but at the cost of poor scalability: each new domain requires re-annotation and feature re-engineering (*Lu et al., 2023*).

In contrast, deep-learning–based IE has demonstrated strong feature-learning capabilities, automatically encoding contextual semantics through neural embeddings and attention mechanisms. Yet, most prior work focuses on well-curated corpora (*e.g.*, newswire) and does not address the cross-site transfer problem essential for web-scale public-opinion mining. Transfer-learning strategies—such as reusing BERT pre-trained on large text corpora and fine-tuning on a small seed set—offer a path to generalization, but their application to noisy web pages remains under-explored (*Wang, 2023*).

To bridge these gaps, we introduce Web Information Extraction Model–Deep Learning (WIEM-DL), an end-to-end framework that:

(1) Extracts fine-grained "knowledge units" (sentiment expressions, opinion holders, targets, media sources, ontology concepts) rather than entire page blocks, boosting precision in complex opinion contexts.

(2) Leverages transfer learning—combining bidirectional encoder representations from Transformers (BERT) embeddings with bilateral long short-term memory (BiLSTM), attention, and conditional random fields (CRF) decoding—to adapt quickly to unseen websites with minimal annotation, addressing the scarcity of labeled data in specialized domains.

(3) Integrates lightweight post-processing to normalize extracted entities and relations into a unified public-opinion ontology, supporting downstream tasks like entity-relation extraction and event detection.

The unique feature of the WIEM-DL information extraction model lies in its design to create a transferable extraction model that can extract the required information from media websites with various structures and integrate it into a unified public opinion semantic knowledge base. The model trains with a small amount of initial labeled data and employs transfer learning techniques to apply the model to websites that were not seen during training, such as $Web_1$, $Web_2$, …$n$ in Fig. 1. These websites contain the desired information but have different webpage structures.

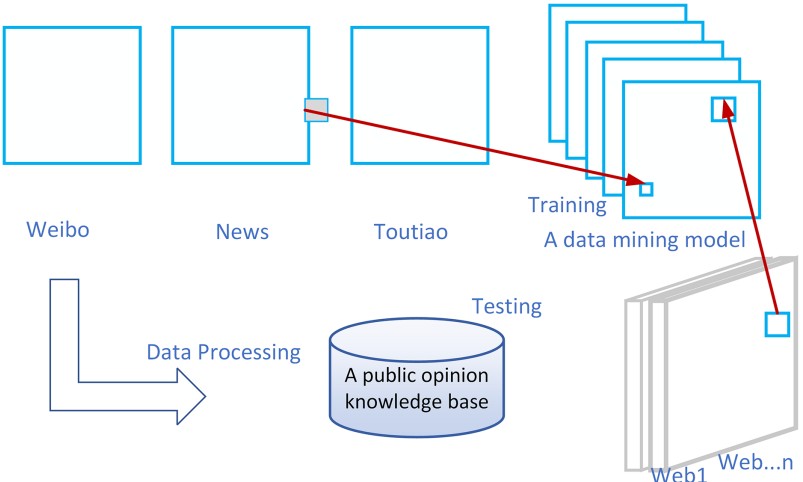

**Figure 1 Description diagram of WIEM-DL public opinion information extraction problem.**

We evaluate WIEM-DL on multiple real-world water-conservancy websites—covering reservoir monitoring forums, environmental news portals, and microblog comment streams—and demonstrate significant improvements in extraction accuracy and throughput over strong baselines (BERT-CRF, BiLSTM-CRF). Our approach reduces manual feature engineering by over 80% while maintaining an F1-score gain of 10–15% in entity and sentiment extraction tasks.

The remainder of this article is organized as follows. "Materials and Methods" reviews related work in DOM-based scraping, deep-learning IE, and transfer learning. "Results" details the WIEM-DL architecture. "Discussion" describes the extraction algorithm and training strategy. "Conclusions" presents experimental setup, results, and ablation studies. The article concludes with future research directions.

## MATERIALS AND METHODS

The Water Conservancy Public-Opinion Knowledge Base is a structured repository designed to collect, organize, and analyze public sentiment and discourse related to water management. By leveraging advances in artificial intelligence (AI)—such as natural language processing (NLP), machine learning, and knowledge-graph techniques—it transforms large, unstructured opinion datasets into actionable insights for forecasting trends, guiding policy decisions, and supporting real-time management. This platform underpins research into scalable, domain-specific public-opinion analytics and enables the rapid integration of emerging web-scale data sources.

The development of a domain-specific semantic knowledge base is essential for effective public opinion analysis, particularly in specialized fields such as water conservancy. Traditional approaches to knowledge base construction—manual, automatic, and open methods—each present limitations in scalability, adaptability, and semantic depth.

(*Ahlawat & Choudhary, 2020*). To address these challenges, the WIEM-DL model integrates ontology-based frameworks and semantic web standards, including resource description framework (RDF), resource description framework schema (RDFS), web ontology language (OWL), and semantic web rule language (SWRL), to facilitate structured and semantically rich representations of public opinion data. This integration enables the model to capture complex relationships and domain-specific nuances effectively (*Sen et al., 2017*). Furthermore, the incorporation of fuzzy ontology techniques allows the WIEM-DL model to handle the inherent ambiguity and uncertainty present in public opinion texts. By leveraging these advanced semantic technologies, the WIEM-DL model enhances the accuracy and relevance of information extraction processes, thereby supporting more informed decision-making in the water conservancy sector (*Ma & Tian, 2020*).

Existing semantic knowledge bases have provided foundational resources for general-purpose NLP and commonsense reasoning. For example, WordNet groups English nouns, verbs, adjectives, and adverbs into synsets linked by semantic relations such as hypernymy and meronymy. MindNet automatically derives lexico-semantic relations from dictionary text and free corpora, yielding a large-scale network of labeled word relations without manual curation. OpenCyc offers an open-source slice of the Cyc ontology, containing hundreds of thousands of commonsense concepts and millions of taxonomic assertions under an Apache license (*Chen et al., 2019*). However, these resources focus primarily on broad, domain-agnostic vocabularies and relations. They lack fine-grained, domain-specific coverage for specialized fields—such as water-conservancy terminology, sentiment-laden expressions in environmental discourse, or institutional actors unique to water-resource management. Consequently, they cannot directly support the highly targeted extraction and reasoning tasks required for building a water-conservancy public-opinion knowledge base, which demands precise ontological classes (*e.g.*, reservoir safety incident, flood mitigation policy) and sentiment concepts (*e.g.*, public concern, regulatory approval) beyond the scope of general-purpose systems. By highlighting this gap, we motivate the development of a tailored, domain-specific semantic knowledge base that integrates deep-learning extraction (WIEM-DL) with a specialized ontology for water-conservancy public opinion.

We introduce WIEM-DL, a transferable deep-learning web information extraction (WIE) model that automatically learns both the structural and semantic extraction rules from diverse websites, enabling zero-shot adaptation to new page layouts.

Traditional rule-based WIE systems discover extraction patterns *via* hand-crafted rules and wrappers, relying on domain experts to define templates for each site. These approaches, while precise in narrow domains, break when HTML structures change and incur high maintenance costs (*Niu & Suen, 2012*). Web mining combines techniques from NLP, AI, information retrieval, and machine learning to analyze semi-structured web data—covering statistical analysis, association-rule mining, clustering, classification, and sequential-pattern discovery (*Miwa & Bansal, 2016*). However, these methods extract patterns at the surface level, lacking deep contextual understanding. Supervised and semi-supervised WIE models (*e.g.*, Web Hosted Information into Summarized Knowledge

(WHISK), information extraction based on pattern discovery (IEPAD)) mitigate manual rule creation by learning from annotated examples, yet still require substantial labeled data and struggle to generalize across heterogeneous sites. In contrast, WIEM-DL leverages transfer learning—initializing from a pre-trained BERT embedding and fine-tuning a BiLSTM + CRF sequence-tagging architecture—to automatically extract both entities and sentiments at the paragraph level, significantly reducing annotation effort for each new website.

In contrast, WIEM-DL leverages transfer learning—initializing from a pre-trained BERT embedding and fine-tuning a BiLSTM + CRF sequence-tagging architecture—to automatically extract both entities and sentiments at the paragraph level, significantly reducing annotation effort for each new website (*Wang et al., 2016*). Finally, lightweight post-processing aligns extracted entities to a domain OWL ontology—using standards like RDF and RDFS—to populate a structured public-opinion knowledge base, supporting downstream analytics and decision-making in water-conservancy management.

Supervised WIE trains extraction models on fully annotated web pages, using systems such as WHISK and DEByE to learn extraction rules directly from examples (*Katiyar & Cardie, 2017*). While it reduces the need for hand-coding wrappers and scales beyond purely rule-based methods, it depends on large volumes of expert annotations, which are costly to produce and difficult to repurpose for open-domain extraction.

Semi-supervised WIE (*e.g.*, IEPAD, OLERA, Thresher) bootstraps from a small labeled seed set and automatically expands patterns to unlabeled pages, cutting annotation requirements by up to 80% (*Zhang et al., 2022*). However, its reliance on domain-specific heuristics still hampers generalization to sites with different HTML structures or topic shifts.

Unsupervised WIE frameworks such as RoadRunner and DEPTA infer page templates and extract data with zero manual labels (*Shao & Cheng, 2021*). These methods excel on highly regular sites but fail when layouts change or content drifts, requiring repeated template re-engineering (*Yan et al., 2023*). Modifying templates requires significant manual intervention, further reducing extraction efficiency.

Across all these approaches, manual intervention—whether in annotation, rule writing, or template design—remains a bottleneck, limiting flexibility, increasing costs, and undermining robustness to heterogeneous, evolving web sources (*Geng et al., 2023*).

Deep learning has revolutionized natural language processing by automatically learning hierarchical feature representations from raw text, overcoming the limited generalization of rule-based systems (*Li et al., 2022*). Architectures such as convolutional neural networks (CNNs) excel at capturing local n-gram features for tasks like text classification, while recurrent neural networks (RNNs), particularly long short-term memory (LSTM) units, model sequential dependencies crucial for parsing and generation.

Named entity recognition (NER) benefits from deep learning's end-to-end pipelines, where word and character embeddings feed into sequence encoders and tag decoders (*Yan et al., 2022*). For example, the BiLSTM-CRF architecture combines bidirectional LSTMs with a conditional random field layer to enforce valid tag sequences, achieving state-of-the-art performance on the CoNLL-2003 benchmark. Extensions incorporating

CNNs for character-level features further improve recognition of rare or out-of-vocabulary entities. Surveys document that these models outperform traditional CRF-only systems by 5–10 F1 points, highlighting their robustness in capturing both local and long-range dependencies (*He, Ma & Wang, 2022*; *Gong et al., 2018*).

Despite these advances, deep-learning IE methods typically require large labeled corpora and struggle to generalize across heterogeneous web layouts (*Sato & Tanigawa, 2005*; *Ahmad & Lee, 2008*). Supervised wrapper induction still demands extensive annotations, while unsupervised template learning breaks under structural variation. Semi-supervised approaches reduce labeling by 50–80% but remain domain-bound due to reliance on prior wrappers (*Liu et al., 2019*). These limitations motivate WIEM-DL, which leverages transfer learning (BERT→BiLSTM→CRF) and self-attention to autonomously adapt to new websites with minimal annotation, bridging the gap between high-performance NER models and practical, scalable web IE across diverse water-conservancy sources.

## The overall structure of the WIEM-DL for public opinion information extraction

Web pages combine unstructured text, semi-structured markup, and rigidly structured elements, resulting in a highly heterogeneous information landscape that defies one-size-fits-all extraction strategies. Traditional extraction pipelines that depend on surface-level feature matching—for example, using pattern templates or similarity metrics—often fail to discern subtle semantic links in short text fragments, introducing noise and misclassification when contextual cues are lacking or ambiguous (*Jiang & Ren, 2022*). To overcome the shortcomings of traditional extraction and sentiment-analysis techniques, we propose WIEM-DL, depicted in Fig. 2. Unlike rule- or dictionary-based methods, WIEM-DL applies a fully contextualized approach: it integrates BERT for deep semantic embeddings, a BiLSTM layer to model sequential dependencies, and a CRF decoder for coherent label sequences. This architecture enables precise detection of emotional expressions—for example, identifying subtle sentiment shifts in short opinion fragments—by learning nuanced context rather than relying on fixed lexicons. Additionally, WIEM-DL incorporates a domain-specific ontology of public-opinion concepts (*e.g.*, stakeholder sentiment, regulatory response, infrastructure risk) that enriches the model's representations and guides its generalization to new water-conservancy sites. Post-extraction, entities and sentiments are mapped to this ontology, ensuring consistency and enabling downstream tasks like relation extraction and event monitoring.

In the WIEM-DL model proposed in Fig. 2, we segment web pages into paragraph-level DOM nodes, treating each node as an independent semantic unit to mitigate noise from boilerplate and mixed content.

This fine-grained segmentation aligns with findings that paragraph-level attention mechanisms significantly boost performance in text segmentation and information extraction tasks by isolating coherent concepts. We then encode each paragraph's text with contextual embeddings (*e.g.*, BERT) and model local dependencies using BiLSTM layers,

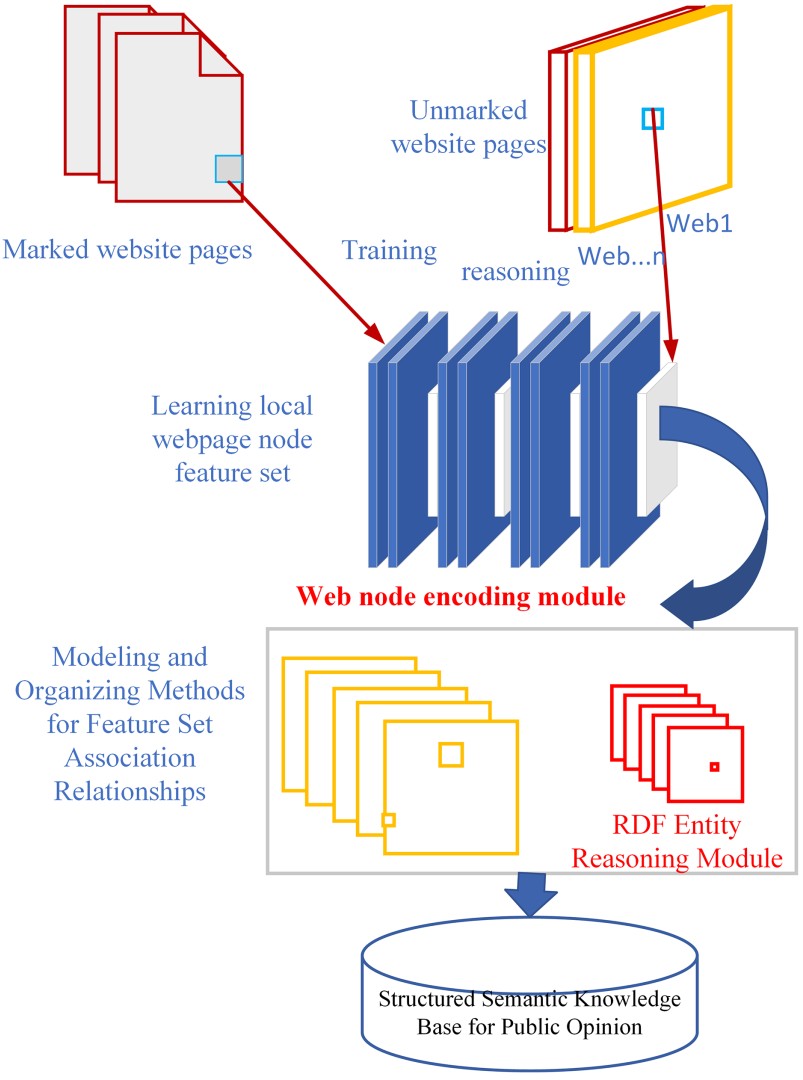

**Figure 2 WIEM-DL model framework description.** 

leveraging the DOM tree structure to capture inter-paragraph relationships and preserve layout semantics.

A self-attention module further highlights sentiment- or entity-rich tokens within each paragraph, enabling more precise extraction of opinion elements compared to global page-level models. Finally, the trained model generalizes to unseen websites—directly applying its learned paragraph-level patterns without wrapper reengineering—thus ensuring robust, scalable extraction across diverse water-conservancy platforms.

In constructing the model, this study leverages the characteristics of web page information by employing word embeddings and a SoftMax layer. The core of the model consists of two RNN layers, designed to learn the features of textual information across web pages. During the process of extracting web page information, the focus is primarily on the text within the nodes themselves, the preceding text of the nodes, and certain discrete information related to the nodes (*Yang et al., 2018*).

To capture the features of the nodes, the model employs dual-path encoding using a word embedding table (WET) and a character embedding table (CET). These tables encode both the sentences and the preceding delimiters, enabling comprehensive modeling of the textual information and generating corresponding feature vectors.

As shown in Fig. 2, the model consists of two key modules: the Web page extraction module for training websites and the information extraction module for testing websites. These two modules form the first two layers of the model and work together to complete the task of extracting information from web pages. The last two layers are constructed based on specific task requirements, and their design and implementation depend on the problem to be solved. With this modular combination, the model can effectively extract the required information from web pages.

The innovations of the WIEM-DL model mainly include the following aspects:

(1) Transferable model: The WIEM-DL model can utilize a small amount of initially labeled website data during the training process and automatically extend to other websites within the same domain. This allows the model to adapt to different website structures and rules, improving its generalization ability and transferability.

(2) Incorporating Sentiment Analysis algorithm: The WIEM-DL model introduces a sentiment analysis algorithm that processes text, speech, and image data to identify the emotions and sentiment tendencies expressed in the information. It classifies and analyzes the sentiment, fully utilizing emotional information in sentiment classification tasks, thereby enabling the model to comprehensively analyze key public opinion content.

(3) Emotion key information extraction based on attention mechanism: The WIEM-DL model employs the attention mechanism to extract key emotional information from public opinion, enhancing its ability to accurately identify sentiment tendencies in the public opinion.

(4) Innovative organizational method: The WIEM-DL model proposes a new method for organizing semantic knowledge in public opinion. By post-processing and integrating the extracted information, a structured public opinion knowledge base is constructed. This organizational method helps better manage and utilize the extracted public opinion information, providing a foundation for subsequent analysis and applications.

## WIEM-DL sentiment information extraction model learning algorithm

After training, the website information extraction module can perform information extraction tasks on different pages of the same website, enabling automatic recognition and extraction of target information. By applying this module across multiple pages of the same website, the efficiency and consistency of information extraction can be improved, facilitating the collection and integration of information from multiple pages within the website. This module helps accelerate the information gathering process, reduces manual operations, and enhances the accuracy and consistency of information extraction (*Wang et al., 2019a*).

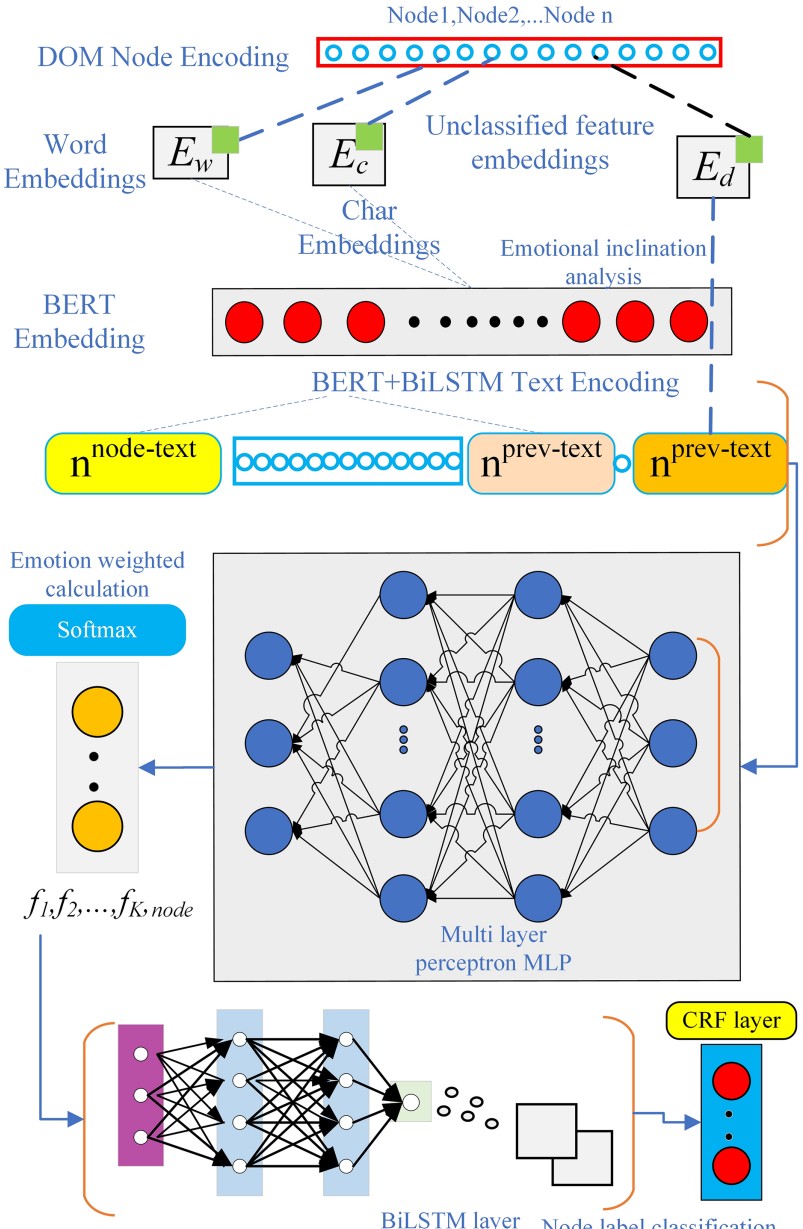

**Figure 3** **WIEM-DL model local node learning structure diagram.**

The website information extraction module is the first stage of the model proposed in this article. The model needs to ensure highly accurate extraction of information from training pages. In Fig. 2, the part that extracts the first stage learning of local webpage node features is shown, and Fig. 3 illustrates the information extraction model for training websites. structure for the text stage. Subsequently, during the process of generating word vectors, word embeddings are applied to each group of text nodes. Next, an RNN is used to construct the target information and the similarity between individual nodes. Another RNN is built to process the node sequence, mainly for calculating the sequential

relationship and position of the target information. This is because, in the process of webpage information extraction, the nodes' positions and their respective order are essential for extracting specific information.

By using the attention mechanism to understand the key emotional information embedded in the text, the model improves the final classification accuracy. When outputting the results, the SoftMax layer computes the classification probabilities, converting the emotion analysis weight vectors into probability distributions.

### Data preprocessing and entity definition

Using web crawlers for data extraction is simple and convenient, but the quality of the data is often low. Therefore, it is necessary to first preprocess the obtained data to remove meaningless parts such as spaces or blank lines. During preprocessing, the text is segmented by sentences, ensuring that no sentence exceeds the maximum input limit of the network. Then, through data annotation, the relationships between the required information and the data are clarified, generating a usable training set.

In the WIEM-DL model, the multilayer perceptron (MLP) plays a crucial role in processing and extracting feature representations from unclassified data. The MLP utilizes layer-by-layer nonlinear transformations to convert input unclassified feature data into higher-level, abstract feature representations. Each layer in the MLP is designed to capture semantic information and patterns at varying levels, gradually extracting more expressive and meaningful features. The final layer of the MLP is a fully connected layer with a SoftMax activation function, which is used for classification and prediction tasks. This layer maps the feature representations to the corresponding sentiment categories or prediction outcomes, outputting a probability distribution *via* the SoftMax function.

In the WIEM-DL model, the Transformer is primarily utilized for semantic modeling and relationship identification of classified sentiment data. By leveraging the self-attention mechanism, the Transformer models relationships and dependencies within the input sequence, effectively capturing long-range dependencies. Through the computation of self-attention weights, the Transformer determines the level of focus on different positions within the input sequence, enabling a more nuanced understanding of semantic relationships.

The identification of node information is the first and a critical step in text information extraction. This process involves detecting the boundaries of phrases corresponding to nodes and determining their node types. For a given sentence composed of words $W: w_1, w_2, \ldots, w_n$ node information recognition assigns a series of labels $N: N_1, N_2, \ldots, N_n$ from a predefined set of categories $N_i \in \phi, |\phi| = k$.

In this study, the Begin, Middle, Other, End, Single (BMOES) labeling scheme is utilized: sentences are input through word embedding, while words are input through character embedding. Here, B-X and M-X indicate that a character is at the beginning or middle of an entity X, O represents a non-entity, E-X indicates the end of entity X, and S-X represents a word that itself forms an entity. The specific definitions of node information labels are detailed in Table 1.

**Table 1  List of semantic knowledge labels for water conservancy public opinion.**

| Tag name | Connotation | Example |
|---|---|---|
| SJ | Public opinion theme | Water pollution fraud incident |
| PT | Publication time | October 11, 2024 |
| TU | Author | Zhang San |
| QXY | Emotional inclination | Positive or negative emotional tendencies |
| CT | Public opinion content | After sorting out, there have been over 11 cases of water pollution fraud that have been investigated and dealt with this year alone |
| RM | Comment | Falsification by water pollution testing agencies: chaos continues, hurting public opinion |
| IP | Reader IP | 134.212.1.1 |
| PER | Personnel involved | Li Si |
| LOC | Involving cities | Yulin |
| NUM | Quantity | 11 |
| RT | Results | Timely identify the cause |
| KW | Keywords | Water resource pollution |

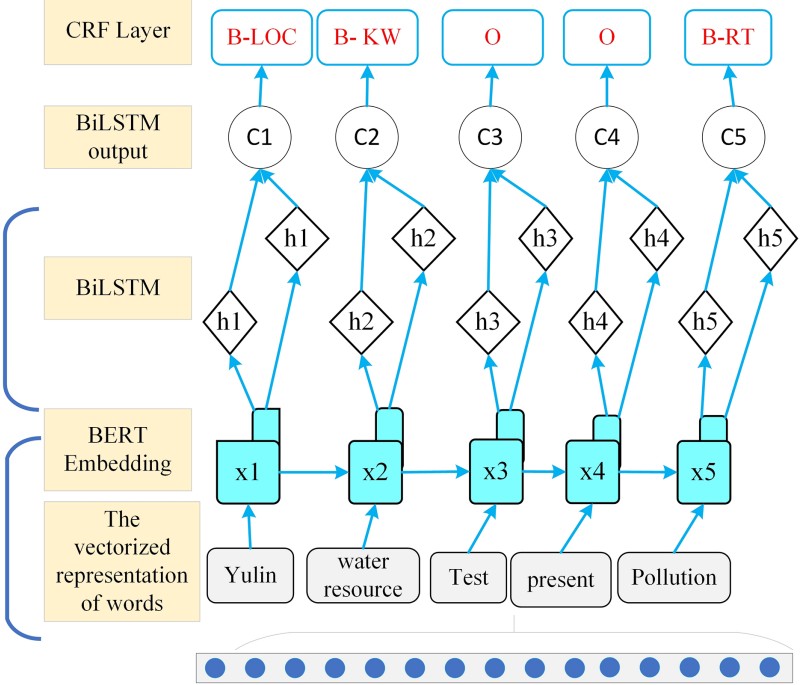

**Figure 4  WIEM-DL model local node learning structure diagram.**

The word segmentation method based on word formation was proposed alongside the development of artificial neural networks. This approach treats Chinese word segmentation as a character labeling problem, utilizing machine learning methods for training to achieve segmentation results. For the sequence decoding method used in node information recognition, the WIEM-DL model employs a localized node learning approach. Its overall framework is illustrated in Fig. 4.

The first step is to map each input $x_i$ to a low-dimensional vector $h_i$ through a feedforward neural network (FNN), $h_i = f(x_i)$, where $f$ is the mapping function of the FNN. Next, calculate the weight $\omega_i$ for each input $x_i$, using the following formula:

$$\omega_i = \frac{\exp(e_i)}{\sum_{j=1}^{n} \exp(e_j)}. \tag{1}$$

In which $e_i$ is a score that measures the correlation between $h_i$ and the entire input sequence. It can be computed using methods such as dot product or a multilayer perceptron (MLP). This article adopts the dot product scoring method, expressed as:

$$e_i = h_i^T \cdot u \tag{2}$$

In which $u$ is a learnable vector.

Finally, the output $y$ is calculated by taking the weighted sum of the inputs $x_i$, as follows:

$$y = \sum_{i=1}^{n} \omega_i \cdot h_i. \tag{3}$$

In which $\omega_i$ is the attention weight assigned to each input $x_i$, and $h_i$ is the transformed low-dimensional vector representation of $x_i$.

In the BERT Embedding layer with integrated sentiment analysis, the attention mechanism can be applied at different levels of public opinion text representation, such as word, sentence, or document level, to capture various levels of sentiment information. The structure of the BERT Embedding layer with integrated sentiment analysis is shown in Fig. 5.

The innovation of the BERT Embedding layer integrated with sentiment analysis lies in its ability to not only use the BERT model as a text representation tool but also introduce elements of sentiment analysis to enhance the model's understanding and processing of emotions. Specifically, this layer builds on BERT by incorporating additional sentiment tags, which strengthens the model's ability to capture emotional information. As a result, the text representation generated by BERT becomes more comprehensive by including sentiment-related features, allowing the model to better understand and classify the emotional tone within the text. This integration helps the model achieve a deeper understanding of both the semantic meaning and the emotional nuances of the text, making it particularly effective for sentiment-sensitive tasks such as public opinion extraction, sentiment classification, and emotion-aware information retrieval. WIEM-DL prepends or appends tokens such as [SENT_POS] and [SENT_NEG] to each paragraph input, guiding BERT's self-attention layers to treat sentiment polarity as an explicit context feature. Sentiment scores from lexicons are concatenated with BERT's wordpiece

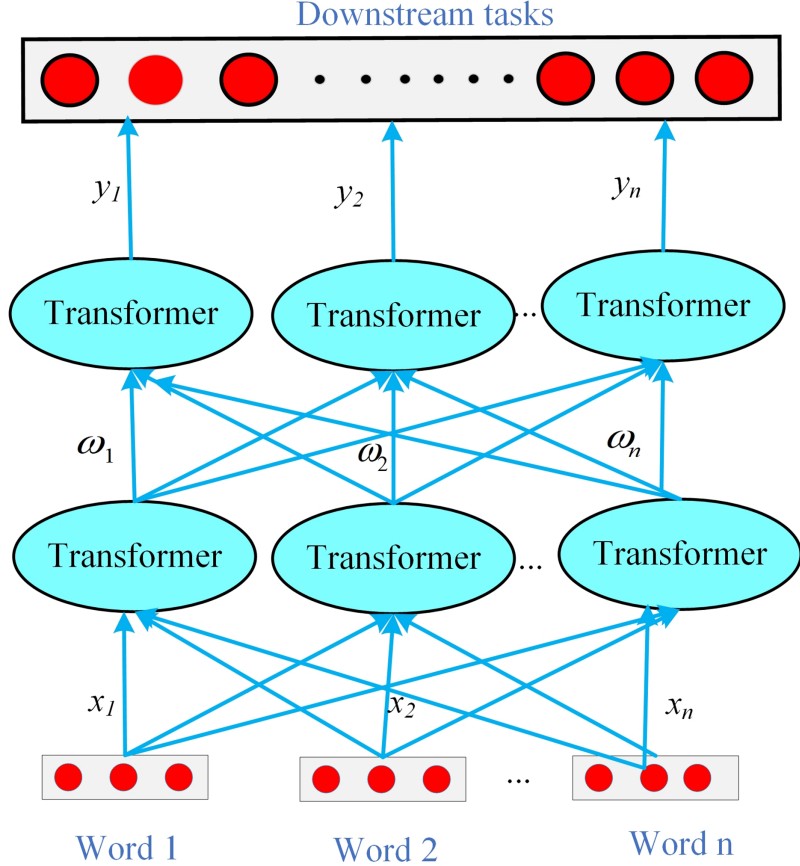

**Figure 5 The BERT Embedding layer integrated with sentiment analysis in the WIEM-DL model.**

embeddings. This fusion of data-driven and lexicon-driven features supplies predefined polarity signals directly into the embedding space. WIEM-DL adds a parallel sentiment classification head alongside the primary NER/extraction head. During fine-tuning, the combined loss $\mathcal{L} = \alpha\mathcal{L}_{extraction} + (1 - \alpha)\mathcal{L}_{sentiment}$ encourages the Transformer layers to internalize both semantic and emotional representations. Inspired by syntax-aware models, WIEM-DL supervises intermediate phrase representations with sentiment labels (positive/negative/neutral). This ensures that embeddings capture compositional sentiment semantics, such as negation or contrast clauses.

### BiLSTM layer of WIEM-DL model

In the WIEM-DL model, the BiLSTM layer is primarily used to process the textual information of each DOM node. Specifically, this layer takes the text representation of each node as input and maps it to a fixed-length vector. The BiLSTM layer is a type of recurrent neural network that processes sequential data by repeatedly applying the same layer at each time step (*Wang et al., 2019b*). The key feature of BiLSTM is its ability to capture both past and future context, as it processes the input sequence in two directions: forward and backward. This allows the model to better understand the semantics of a node by

leveraging the context of surrounding nodes, rather than just focusing on the node's own textual information. By incorporating this bidirectional context, the BiLSTM layer enables the model to generate more accurate and comprehensive semantic representations of the text, which is essential for tasks such as information extraction and sentiment analysis in the context of web pages.

In the WIEM-DL model, the BiLSTM layer consists of two LSTM layers: one is the forward LSTM, and the other is the backward LSTM. Both LSTM layers share the same parameters, but they process the input sequence in opposite directions. The forward LSTM processes the sequence from left to right, while the backward LSTM processes the sequence from right to left.

This bidirectional structure allows the model to simultaneously consider both the forward and backward context of each node, capturing the information from both past and future nodes in the sequence. By integrating these two LSTM layers, the BiLSTM layer can generate more comprehensive and context-aware representations, which is crucial for understanding the full semantic meaning of the node's text and improving the performance of tasks such as information extraction and sentiment analysis.

The output of the BiLSTM layer is a sequence of hidden states for each node, where each hidden state is a vector representing the semantic information of the node. These vectors can then be input into the attention mechanism to compute the importance weight of each node, helping the model better learn the semantic information of key nodes.

The model's structure includes components such as memory cells, input gates, forget gates, and output gates. These components form a control mechanism that significantly enhances the model's ability to capture long-range historical information. In the WIEM-DL model, the BiLSTM layer processes the input vectors generated by the BERT Embedding layer. The vector sequence is split into forward and backward LSTMs, and the outputs from these two LSTMs are concatenated in the hidden layer to form a BiLSTM. The final output layer consists of a fully connected layer, and the length of the output vector corresponds to the number of labels in the task.

This design allows the BiLSTM layer to effectively capture both past and future context of the input nodes, enabling more accurate modeling of the semantic information needed for tasks such as sentiment analysis and information extraction. Let the sequence $X = (x_1, x_2, \ldots, x_s)$ represent the input sequence, where $s$ denotes the sequence length. In this article, for each word in the sentence, an embedding vector $i = [w; c_h; c_a]$ is constructed, consisting of three parts: the semantic-level features $w$, the character-level features $c_h$, and the case-sensitive features $c_a$. The dimensions of these three parts are denoted as $d_w$, $d_o$ and $d_c$.

In which $d_w$ represents the dimension of the semantic-level features $w$, $d_o$ refers to the dimension of the output-level features $c_h$ , $d_c$ is the dimension of the character-level features $c_a$. Each of these components contributes a specific aspect to the word's embedding, allowing the model to learn richer and more nuanced representations of the input sequence. In order to obtain comprehensive and complete feature information, character-level features $c_h$ are introduced. Specifically, the BERT Embedding model is used to input the character embeddings, weighted by emotional attention, into an LSTM. The $n$ output vector of the LSTM represents the character-level feature $C_h$. This approach allows

the model to capture finer-grained information at the character level, which is particularly useful for tasks such as handling out-of-vocabulary words or understanding the morphological structure of words. By integrating this character-level information with the semantic-level features from BERT, the model can create a more robust and nuanced representation of the input sequence. Where the character length of a word is denoted by $n$, so the total length of the input vector is given by $d_i = d_w + d_o + d_c$. The detailed formulas for the LSTM are as follows:

$$i_t = \sigma(w_{xi}x_t + W_{hi}h_{t-1} + W_{ci}c_{t-1} + b_i) \tag{4}$$

$$f_t = \sigma\left(w_{fi}x_t + W_{hf}h_{t-1} + W_{cf}c_{t-1} + b_f\right) \tag{5}$$

$$o_t = f_t c_{t-1} + i_t \tanh(W_{xc}x_t + W_{hc}h_{t-1} + b_c) \tag{6}$$

$$c_t = \sigma(w_{xo}x_t + W_{ho}h_{t-1} + W_{co}c_{t-1} + b_o) \tag{7}$$

$$h_t = o_t \tanh h(c_t). \tag{8}$$

In which $i_t, f_t, o_t, c_t$ represent the input gate, forget gate, output gate, and memory cell at time $t$, respectively. The computations for these gates involve weight matrices $W_x, W_h, W_c$, and bias terms $b_i, b_f, b_c, b_o$. The activation function is denoted by $\sigma$. All these parameters are updated through network training to determine their final values. To simplify the notation, the LSTM model will be represented as LSTM $(i_t, h_{t-1})$ in the subsequent text.

The BiLSTM layer is a deep learning model commonly used for processing sequential data, achieving excellent results in natural language processing tasks. Compared to traditional unidirectional LSTM, BiLSTM considers the contextual information for each word, utilizing the words on both the left and right sides of the current word to predict its label. Specifically, BiLSTM consists of a forward LSTM and a backward LSTM, which compute the hidden states of each word from left to right and from right to left, respectively. The two hidden states are then concatenated to form the vector representation of the current word. This approach enables the model to capture contextual information more effectively (*Wang, Xue & Zhang, 2021*). In named entity recognition (NER) tasks, a BiLSTM can pass the vector output of each word as input to a conditional random field (CRF) layer to achieve more accurate sequence labeling. The CRF layer models all possible labeling sequences by calculating scores for each potential sequence based on the given input. It then outputs the labeling sequence with the highest probability. During training, the CRF layer uses the backpropagation algorithm to update model parameters, minimizing the gap between the model output and the ground truth labels.

The BiLSTM layers in the WIEM-DL model can be stacked to further enhance the model's performance. Specifically, in the WIEM-DL model, the BiLSTM consists of two stacked layers, each containing 128 hidden units. These hidden units capture the relationships between the current word and its surrounding context, generating vector representations of the words for subsequent use by the CRF layer (*Zhang, Xue & Zhang, 2019*). The primary purpose of the LSTM model is to extract latent features from sequences. The extraction model can obtain embeddings for all words in the embedding layer, and these embeddings are ordered to form a sequence $I = (i_1, \ldots, i_t, \ldots, i_s)$, which serves as the input to the LSTM, where $i_t \in \mathbb{R}^{dt}$. In the LSTM model, there are two separate LSTM modules: one operates forward and the other backward, extracting hidden

features $\overrightarrow{h_t}$ and $\overleftarrow{h_t}$, respectively. The bidirectional LSTM can be described using the following equations:

$$\overrightarrow{h_t}, \overrightarrow{c_t} = LSTM(i_t, \overrightarrow{h}_{t-1}, \overrightarrow{c}_{t-1}), \overleftarrow{h_t}, \overleftarrow{c_t} = LSTM(i_t, \overleftarrow{h}_{t-1}, \overleftarrow{c}_{t-1}). \tag{9}$$

The output of the LSTM is a concatenation of the forward and backward hidden layer features $h_t = \left[\overrightarrow{h_t}, \overleftarrow{h_t}\right]$. Subsequently, an activation function is applied to extract features from $\overrightarrow{h_t}$ and $\overleftarrow{h_t}$, and a linear layer maps the hidden features to $d_l$ dimensional space. Where $d_l$ represents the number of entity types to be recognized. $W_l$ and $W_t$ are the feature matrices, and $b_t$ and $b_l$ are the bias terms. Since the BiLSTM layer can capture the contextual information of words in a sentence, it performs exceptionally well in sentiment analysis. The innovation of the BiLSTM layer in the WIEM-DL model lies primarily in its use of a bidirectional LSTM structure, coupled with the integration of an attention mechanism to enhance the precision of sentiment classification.

By leveraging the bidirectional flow of information, the BiLSTM layer ensures that both preceding and succeeding contexts are considered. The attention mechanism further refines this process by focusing on the most relevant parts of the input sequence, dynamically weighting the importance of words based on their contribution to the sentiment classification task. This combination allows the WIEM-DL model to achieve a deeper understanding of the emotional and semantic nuances present in the text. Traditional LSTM considers only unidirectional information flow, focusing on the influence of preceding words on the current word. In contrast, BiLSTM incorporates bidirectional information flow, capturing contextual information from both preceding and succeeding words. This bidirectional structure enables the model to more accurately grasp key semantic information critical to sentiment analysis.

The attention mechanism enhances this capability by learning the significance of different parts of the text and applying weighted calculations, improving the precision of sentiment classification. It dynamically assigns weights to various positions in the input sequence, allowing the model to focus on information most relevant to the classification task. This approach enhances the model's robustness and generalization ability. Additionally, the BiLSTM layer in the WIEM-DL model integrates batch normalization, which accelerates training, mitigates issues such as vanishing and exploding gradients, and contributes to the model's overall stability and performance.

### CRF layer in WIEM-DL model

Under the condition of a given set of input random variables, the CRF layer models the conditional probability distribution of the output random variables. In the WIEM-DL model, the CRF layer is used for sequence labeling of sentiment labels in text sequences. This layer takes the output of the BiLSTM layer as input and uses a series of parameterized functions to map the feature vectors at each position in the input sequence to corresponding label scores. The CRF layer then performs global normalization to calculate the probability of each label sequence and optimizes the label sequence based on these probabilities, ensuring the best match with the observed sequence. Additionally, the CRF

layer incorporates an attention mechanism, which automatically learns the important information in the text, thereby improving the accuracy of sentiment analysis and event extraction. Specifically, the CRF layer effectively captures the contextual information within a sequence because it considers the entire label sequence simultaneously. Compared to traditional classifiers, the CRF layer can model the dependencies between labels while ensuring the relevance of the output labels, thereby improving the accuracy of sentiment classification. In the WIEM-DL model, the combination of the CRF layer and the BiLSTM layer achieves better performance in sentiment classification tasks compared to traditional methods (*Zhang, Xue & Zhang, 2018*). The linear-chain CRF is a commonly used model in sequence labeling problems, as shown in the brief structural diagram in Fig. 6. In sequence labeling tasks, the labeled sequence and the sequence to be labeled correspond to the state sequence variables $x$ and $y$, respectively.

Given a known training dataset, the model can derive a conditional probability model for the training data using the maximum likelihood estimation (MLE) algorithm. When given an input sequence y to label new data, the model determines the output $x^*$ by maximizing the conditional probability $P(x|y)$, where $x^*$ represents the model output that maximizes the conditional probability. CRF outperform other models in capturing the dependencies between labels, which improves the accuracy of entity recognition. Specifically, in the model, the output of the LSTM layer is combined into a sequence $L = [l_1, l_2, \ldots, l_{s-1}, l_s]^T$, which serves as the input to the CRF model. Where $L$ has dimensions of $s \times d_l$, $L_{i,j}$ represents the probability that the $i$ word in the sentence is predicted to be labeled as the $j$ tag. For a predicted sequence $y = (y_1, y_2, \ldots, y_{s-1}, y_s)$, the score function of the CRF model is defined as follows:

$$f(X, y) = \sum_{i=0}^{s+1} T_{yi,yi+1} + \sum_{i=1}^{s} L_{i,yi}. \tag{10}$$

The transition matrix $T$ is used to represent the transition probabilities between labels, where $T_{i,j}$ denotes the probability of transitioning from the $i$ label to the $j$ label. The labels for the start and end positions are represented by $y_o$ and $y_{s+1}$, respectively. The final output sequence y probability is computed by the SoftMax normalization function in the joint extraction model.

$$p(y|x) = \frac{1}{Z(x)} \exp\left(\sum_{i=1}^{T} \sum_{k=1}^{K} w_k f_k(y_{t-1}, y_t, x, t)\right). \tag{11}$$

In which $Z(x)$ is the normalization factor, which ensures that the sum of the conditional probabilities of all possible label sequences equals 1. $w_k$ is the weight of the feature function $f_k$, where $f_k(y_{t-1}, y_t, x, t)$ represents the combination of features at time step $t$, involving the label $y_t$, its previous label $y_{t-1}$, and the input $x$. The model parameters $w_k$ can be optimized using algorithms such as stochastic gradient descent. The final trained model can then be used to predict new label sequences by solving the following:

$$\hat{y} = argmax_y(p(x|X)). \tag{12}$$

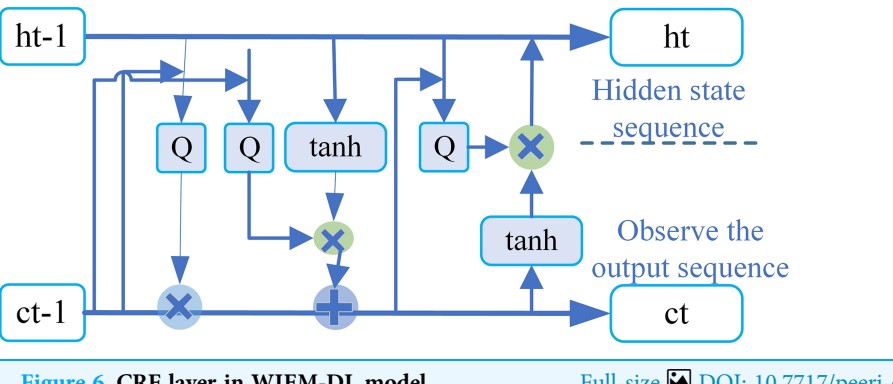

**Figure 6  CRF layer in WIEM-DL model.**

In which $\hat{y}$ represents the label sequence that maximizes the conditional probability $p(x|X)$, where xxx is the input sequence and $x$ represents the model parameters.

The CRF layer in the WIEM-DL model is applied in the field of online public opinion analysis. It performs sentiment analysis and event extraction on online texts, enabling real-time monitoring and analysis of large-scale data from platforms such as social media. This helps users stay informed about the public's attitudes and emotional tendencies toward events and topics. The detailed description of the WIEM-DL algorithm (Algorithm 1) is as follows.

# RESULTS

## Experimental data and preprocessing

This section evaluates the WIEM-DL model and compares it with other well-established extraction models such as BERT-CRF and BiLSTM-CRF, which have demonstrated strong extraction performance. The proposed WIEM-DL model is capable of handling both Chinese and English text. The design and implementation of the model took into account the specific characteristics and processing requirements of both languages. Universal natural language processing techniques and methods, such as word vector representations, sequence labeling models, and semantic relationship extraction, were employed to ensure compatibility with both Chinese and English. Through data preprocessing and model design, the WIEM-DL model can effectively process text in both languages.

The WIEM-DL is an online system that operates on the Windows 10 operating system and uses an Intel(R) Core(TM) i7 3.0GHz processor. In the experiment, a custom Python web crawler was used to collect data from 20,000 web pages, including data from websites such as Weibo, Sina News, and Toutiao. The extracted information includes public opinion topics, sentiment analysis, publication time, author, content, comments, involved people, cities, quantity, results, keywords, and sources.

The experiment initially involved collecting data from Weibo, Sina News, and Toutiao for entity recognition, and then training on other website content. In the experiment, 20,000 manually annotated public opinion pages from Weibo, Sina News, and Toutiao were used as the training set. After training, the model was tasked with extracting target information from 800 other test pages from the same websites. Additionally, some relevant

| **Algorithm 1   WIEM-DL.** |
| --- |

Input: Text data such as web pages, social media content

Output: Sentiment or event recognition results M

1: // Data Collection and Preprocessing

2: Collect web data from various sources (*e.g.*, Weibo, Sina News, Toutiao) using a custom Python web scraper

3: Extract key information from the collected data:

    1. Sentiment analysis (positive/negative/neutral)

    2. Entity recognition (themes, time of publication, author, content, comments, involved persons, city, *etc.*)

4: Preprocess the data to create labeled training sets and extract target information

5: Split data into training set, validation set, and test set (with dev_split_size = 0.1)

6: // Convolutional Module

7: for i = 1 to n do

8:    if i == 1 then

9:        // First convolutional layer

10:      Fi = conv(I, 1D, filters=32)

11:  else

12:        // Residual module for deeper layers

13:       Fi = residual(Fi-1)

14:    end if

15: end for

16: // BiLSTM Layer

17: Fi_bilstm = BiLSTM(Fi) // Process features with BiLSTM

18: // Attention Mechanism

19: Attention_weights = attention(Fi_bilstm)

20: Fi_attention = Fi_bilstm * Attention_weights // Apply attention

21: // CRF Layer

22: CRF_output = CRF(Fi_attention) // Perform sequence labeling with CRF

23: // Final Output Prediction

24: M = softmax(CRF_output) // Predict the final

25: Output the classification result M obtained after the softmax layer operation

public opinion corpora and information extraction datasets for the public opinion domain were gathered for testing and validation. The dataset is shown in Table 2. The initial dataset was divided based on the source platforms—Weibo, Sina News, and Toutiao—to capture the unique characteristics and content styles of each platform. Within each platform, pages covering a range of public opinion topics were identified. This included subjects such as policy discussions, social events, and public reactions to news, ensuring that the training data encompassed various domains of public discourse. To train the model effectively on different emotional tones, pages were selected to represent a balance

of positive, negative, and neutral sentiments. Preliminary sentiment analysis tools were employed to categorize the sentiment polarity of potential pages. From the stratified and sentiment-balanced pool, 20,000 pages were randomly selected for manual annotation. Expert annotators labeled entities and sentiments according to predefined guidelines, ensuring high-quality annotations for model training.

### Evaluation indicators

The text classification extraction model uses accuracy (*Precision*), recall (*Recall*), and F-measure (*F1*) to evaluate the performance of webpage content extraction. The evaluation formulas are as follows:

$$Pre = \frac{|S_e \cap S_t|}{|S_e|} \tag{13}$$

$$Rec = \frac{|S_e \cap S_t|}{|S_t|} \tag{14}$$

$$F1 = \frac{2 * Pre * Rec}{Pre + Rec}. \tag{15}$$

In which $S_e$ represents the extracted results, and $S_t$ represents the manually annotated results.

### Model implementation and training parameter settings

In the Python programming environment, the algorithm was implemented using the Torch framework. The dataset consists of 5,000 sentences, with the training and validation sets split at a ratio of dev_split_size = 0.1. The maximum vocabulary size is set to max_vocab_size = 1,000,000, and the maximum sentence length is limited to max_len = 500. The learning rate is set to learning_rate = 1e−5, with a weight decay of weight_decay = 0.01. Gradient clipping is applied with a value of clip_grad = 5. The batch size is batch_size = 4, and the model is trained for epoch_num = 20 epochs, with an early stopping patience of patience_num = 4. The Adam optimizer was chosen, and the model was trained on the CPU for 20 epochs. After completing the model training, the model can be applied to web information extraction tasks. By using this model, we can extract target data from web pages and generate corresponding entity information. The application of these entity information can be referenced in Fig. 2, which demonstrates an example of entity information generated by the extraction model. In this way, unstructured web information can be transformed into structured entity data, achieving the integration between structured and unstructured data. Below is a detailed justification for each hyperparameter choice:

(1) A learning rate of 1e−5 is commonly recommended for fine-tuning large pre-trained models like BERT, especially when dealing with smaller datasets. This conservative rate helps prevent catastrophic forgetting and ensures stable convergence during training.

(2) Applying a weight decay of 0.01 serves as a regularization technique to prevent overfitting by penalizing large weights. This value is standard in fine-tuning scenarios and aligns with practices observed in training Transformer models.

**Table 2 Experimental training dataset.**

| Dataset name | Website address | Number of samples collected |
|---|---|---|
| Weibo | http://weibo.com/ | 5,000 |
| Sina News | http://www.sina.com.cn | 5,000 |
| Toutiao | http://toutiao.com/ | 1,000 |
| Weibo-COV (*Ma & Tian, 2020*) | A continuously maintained public opinion *corpus* | 4,099 |
| Weibo NER (*Chen et al., 2019*) | A Chinese NER dataset | 12,259 |

(3) Gradient clipping at a norm of 5.0 is employed to mitigate the risk of exploding gradients, which can destabilize training. This technique is particularly beneficial when training deep networks or when using small batch sizes.

(4) A batch size of 4 is chosen to accommodate memory limitations inherent in CPU-only training environments. While smaller batch sizes can introduce more noise into the training process, they are often necessary when computational resources are limited.

(5) Setting the maximum sentence length to 500 tokens allows the model to capture sufficient context from web page paragraphs, which often contain longer sequences of text. This length balances the need for contextual information with the computational constraints of the training environment.

(6) A large vocabulary size of 1,000,000 ensures comprehensive coverage of diverse terms found in web data, including domain-specific jargon and colloquial expressions. This extensive vocabulary aids in accurate tokenization and representation of input text.

(7) Allocating 10% of the dataset for validation is a standard practice that provides a sufficient sample to evaluate model performance and generalization without significantly reducing the training data size.

(8) Training the model for up to 20 epochs with an early stopping patience of four epochs helps prevent overfitting. Early stopping monitors validation performance and halts training when improvements plateau, which is especially important when working with limited data.

(9) The Adam optimizer is selected for its adaptive learning rate capabilities and efficiency in handling sparse gradients, making it well-suited for training deep learning models like BERT.

The hyperparameter choices for the WIEM-DL model are informed by best practices in training Transformer-based models under resource constraints. These settings aim to balance model performance with computational efficiency, ensuring effective learning from the available data.

### Comparative analysis of experimental results

The comparison methods in this study are BERT-CRF (*Ma & Tian, 2020*) and BiLSTM-CRF (*Chen et al., 2019*) models. The experimental performance of information extraction is evaluated using three metrics: precision (P), recall (R), and F1 score (F1), to verify the effectiveness of the proposed model. In the experiment, 10 manually labeled

news pages, sourced from three different media platforms—Sina Weibo, Sina News, and Toutiao—are used as the training set. The other 20,000 pages from the same websites are used as the test set to validate the model's performance. The overall results of the experiment are shown in Tables 2–4. Table 2 is experimental training dataset,this table outlines the composition of the training dataset used for the WIEM-DL model. Table 3 is training results of information extraction models for three types of websites, this table presents the performance metrics of the WIEM-DL model compared to baseline models across the three platforms. Table 4 is transfer training results of information extraction models for two types of websites, this table evaluates the transfer learning capabilities of the WIEM-DL model when trained on one platform and tested on another.

While the WIEM-DL model demonstrates superior performance over baseline models like BiLSTM-CRF and BERT-CRF, it's crucial to ascertain whether these improvements are statistically significant. Incorporating confidence intervals (CIs) for evaluation metrics such as precision, recall, and F1 score can provide insights into the reliability of the results. For instance, employing bootstrapping methods to compute 95% CIs can offer a clearer picture of the model's performance variability. This approach involves resampling the test data multiple times and calculating the metrics for each sample, thus estimating the range within which the true performance metrics lie with 95% confidence.

From the analysis of the experimental results, we can conclude that the WIEM-DL model achieves an overall extraction accuracy of over 92% for web pages, outperforming both BiLSTM-CRF and BERT-CRF. Specifically, the F1 score for BiLSTM-CRF is 90.64%, while that for BERT-CRF is 89.69%. Except for a few pages with special text formats, the model successfully extracts information from the majority of pages. The detailed performance can be seen in Table 3.

(1) In terms of the F1 evaluation metric, it is evident that the proposed WIEM-DL model achieves the best performance on most datasets. Notably, the model attains the highest F1 score of 95% on the dataset sourced from Sina News. The average F1 score across all datasets exceeds 93%. These experimental results demonstrate that the comprehensive capability of WIEM-DL significantly surpasses that of the other comparative methods.

(2) By comparing the Weibo and news datasets, it can be observed that datasets with longer content tend to achieve higher F1 scores. Unlike the news dataset, the Weibo dataset primarily consists of short texts with relatively higher noise levels. Training the model on such datasets makes it challenging for the model to accurately pinpoint the relevant content, which subsequently reduces its performance.

(3) In comparison with the BERT-CRF and BiLSTM-CRF models, the WIEM-DL model demonstrates superior performance in completing information extraction tasks regardless of text length. It consistently retrieves accurate content information. These comparative results validate the effectiveness of the proposed model and highlight that leveraging title-based content localization can significantly enhance the accuracy and robustness of extraction models.

(4) Finally, analyzing the experimental results from the perspective of accuracy, the WIEM-DL model achieved the highest accuracy compared to the BERT-CRF and

**Table 3 Training results of information extraction models for three types of websites.**

| Label source | Model | Indicators (%) | SJ | PT | TU | CT | RM | PER | QXY | KW | Average value |
|---|---|---|---|---|---|---|---|---|---|---|---|
| Sina Weibo | BERT-CRF | Pre | 0.84 | 0.91 | 0.95 | 0.99 | 0.87 | 0.88 | 0.86 | 0.95 | 0.91 |
| | | Rec | 0.89 | 0.94 | 0.87 | 0.78 | 0.92 | 0.88 | 0.85 | 0.78 | 0.87 |
| | | F1 | 0.86 | 0.92 | 0.91 | 0.87 | 0.89 | 0.88 | 0.85 | 0.86 | 0.88 |
| | BiLSTM-CRF | Pre | 0.75 | 0.86 | 0.94 | 0.86 | 0.96 | 0.75 | 0.98 | 0.96 | 0.89 |
| | | Rec | 0.85 | 0.77 | 0.75 | 0.86 | 0.93 | 0.81 | 0.94 | 0.97 | 0.86 |
| | | F1 | 0.80 | 0.81 | 0.83 | 0.86 | 0.94 | 0.78 | 0.96 | 0.96 | 0.87 |
| | **WIEM-DL** | Pre | 0.85 | 0.90 | 0.99 | 0.94 | 0.92 | 0.91 | 0.99 | 0.94 | 0.93 |
| | | Rec | 0.97 | 0.97 | 0.90 | 0.89 | 0.90 | 0.94 | 0.89 | 0.89 | 0.92 |
| | | F1 | 0.91 | 0.93 | 0.94 | 0.91 | 0.91 | 0.92 | 0.94 | 0.91 | 0.93 |
| Sina news | BERT-CRF | Pre | 0.84 | 0.82 | 0.79 | 0.94 | 0.89 | 0.93 | 0.82 | 0.82 | 0.86 |
| | | Rec | 0.89 | 0.85 | 0.78 | 0.98 | 0.88 | 0.91 | 0.97 | 0.95 | 0.91 |
| | | F1 | 0.86 | 0.83 | 0.78 | 0.96 | 0.88 | 0.92 | 0.89 | 0.88 | 0.88 |
| | BiLSTM-CRF | Pre | 0.77 | 0.75 | 0.87 | 0.80 | 0.81 | 0.82 | 0.80 | 0.78 | 0.80 |
| | | Rec | 0.90 | 0.79 | 0.87 | 0.94 | 0.91 | 0.82 | 0.79 | 0.78 | 0.85 |
| | | F1 | 0.83 | 0.77 | 0.87 | 0.86 | 0.86 | 0.82 | 0.79 | 0.78 | 0.83 |
| | **WIEM-DL** | Pre | 0.95 | 0.94 | 0.90 | 0.91 | 0.90 | 0.98 | 0.88 | 0.88 | 0.92 |
| | | Rec | 0.96 | 0.93 | 0.98 | 0.98 | 0.89 | 0.90 | 0.95 | 0.92 | 0.94 |
| | | F1 | 0.95 | 0.93 | 0.94 | 0.94 | 0.89 | 0.94 | 0.91 | 0.90 | 0.93 |
| Toutiao | BERT-CRF | Pre | 0.81 | 0.87 | 0.98 | 0.90 | 0.79 | 0.98 | 0.91 | 0.97 | 0.91 |
| | | Rec | 0.98 | 0.76 | 0.98 | 0.99 | 0.89 | 0.98 | 0.84 | 0.89 | 0.92 |
| | | F1 | 0.89 | 0.81 | 0.98 | 0.94 | 0.84 | 0.98 | 0.87 | 0.93 | 0.91 |
| | BiLSTM-CRF | Pre | 0.92 | 0.75 | 0.89 | 0.97 | 0.86 | 0.99 | 0.89 | 0.90 | 0.90 |
| | | Rec | 0.82 | 0.79 | 0.78 | 0.80 | 0.78 | 0.89 | 0.80 | 0.87 | 0.82 |
| | | F1 | 0.87 | 0.77 | 0.83 | 0.88 | 0.82 | 0.94 | 0.84 | 0.88 | 0.86 |
| | **WIEM-DL** | Pre | 0.93 | 0.98 | 0.88 | 0.89 | 0.90 | 0.97 | 0.85 | 0.94 | 0.92 |
| | | Rec | 0.93 | 0.93 | 0.90 | 0.91 | 0.96 | 0.85 | 0.96 | 0.89 | 0.92 |
| | | F1 | 0.93 | 0.95 | 0.89 | 0.90 | 0.93 | 0.91 | 0.90 | 0.91 | 0.92 |

**Table 4 Transfer training results of information extraction models for two types of websites.**

| Label source | Indicators (%) | SJ | PT | TU | CT | RM | PER | LOC | KW | Average value |
|---|---|---|---|---|---|---|---|---|---|---|
| WangYi163 | Pre | 0.90 | 0.93 | 0.97 | 0.90 | 0.90 | 0.94 | 0.93 | 0.99 | 0.94 |
| | Rec | 0.97 | 0.95 | 0.95 | 0.93 | 0.98 | 0.99 | 0.89 | 0.89 | 0.96 |
| | F1 | 0.94 | 0.94 | 0.96 | 0.92 | 0.94 | 0.97 | 0.91 | 0.94 | 0.94 |
| FengHuang | Pre | 0.98 | 0.90 | 0.94 | 0.86 | 0.90 | 0.97 | 0.88 | 0.91 | 0.92 |
| | Rec | 0.81 | 0.95 | 0.87 | 0.94 | 0.85 | 0.90 | 0.94 | 0.86 | 0.89 |
| | F1 | 0.89 | 0.93 | 0.91 | 0.90 | 0.88 | 0.94 | 0.91 | 0.89 | 0.91 |

BiLSTM-CRF models. Specifically, it achieved an accuracy of 93% on the Sina Weibo and Sina News datasets. This high level of accuracy demonstrates the proposed model's strong adaptability and its potential for superior performance in open information extraction tasks.

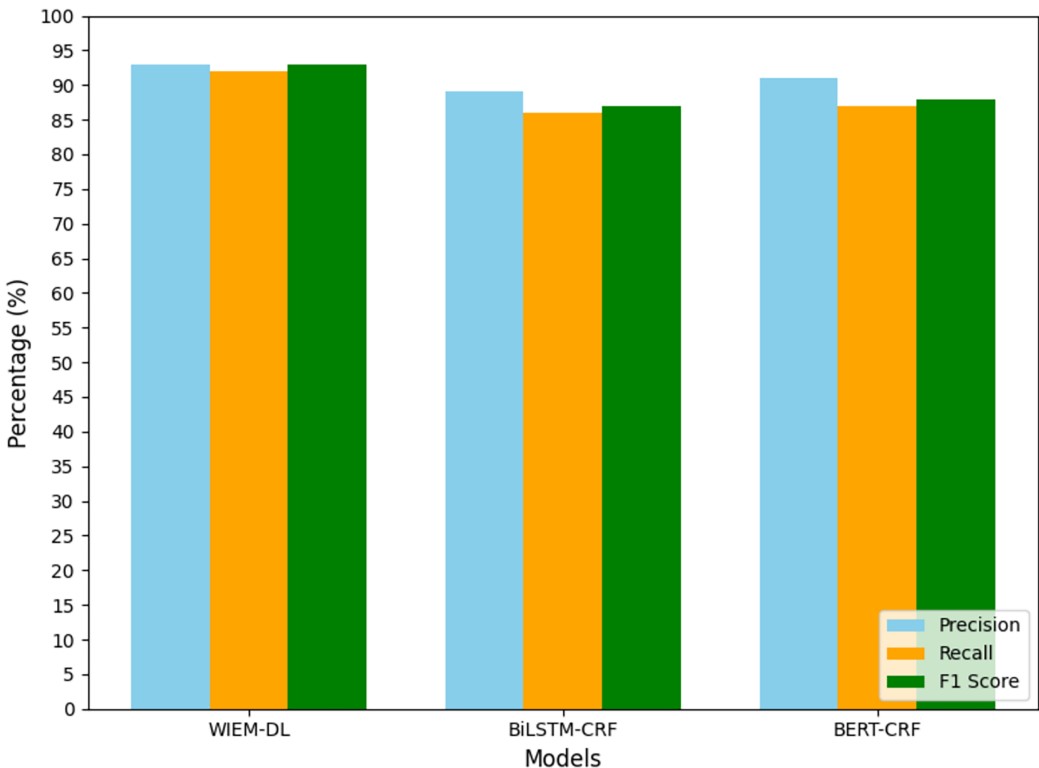

**Figure 7 Statistical chart of page information extraction results on Sina Weibo.**

While WIEM-DL integrates components like BERT embeddings, BiLSTM, and CRF layers, similar architectures have been explored in prior research. Emphasize any novel aspects of the model, such as innovative integration techniques, unique training strategies, or application to new domains. Compare WIEM-DL's performance with a wider array of models, including recent state-of-the-art approaches, to contextualize its effectiveness. Conducting ablation studies is vital to understand the contribution of each component within the WIEM-DL architecture. By systematically removing or altering components like the BERT embeddings, BiLSTM layer, or CRF layer, and observing the impact on performance, one can identify which elements are most critical to the model's success.

To isolate the impact of WIEM-DL's components—BERT embeddings, sentiment-aware tagging, BiLSTM, attention, CRF decoding, and ontology integration—we propose systematic ablations. Each study variant removes or replaces one element, measuring resultant drops in precision, recall, and F1. Paired bootstrap tests and 95% confidence intervals will assess statistical significance. This approach follows established ablation methodologies in NLP and information extraction research. Experimental setup as:

(1) Use the established 20,000-page test set, retaining the original train/validation split.
(2) For each variant, compute precision, recall, and F1, employ paired bootstrap (10,000 resamples) to derive 95% confidence intervals for F1, Apply McNemar's test for pairwise significance ($\alpha = 0.05$) between Full WIEM-DL and each ablation.
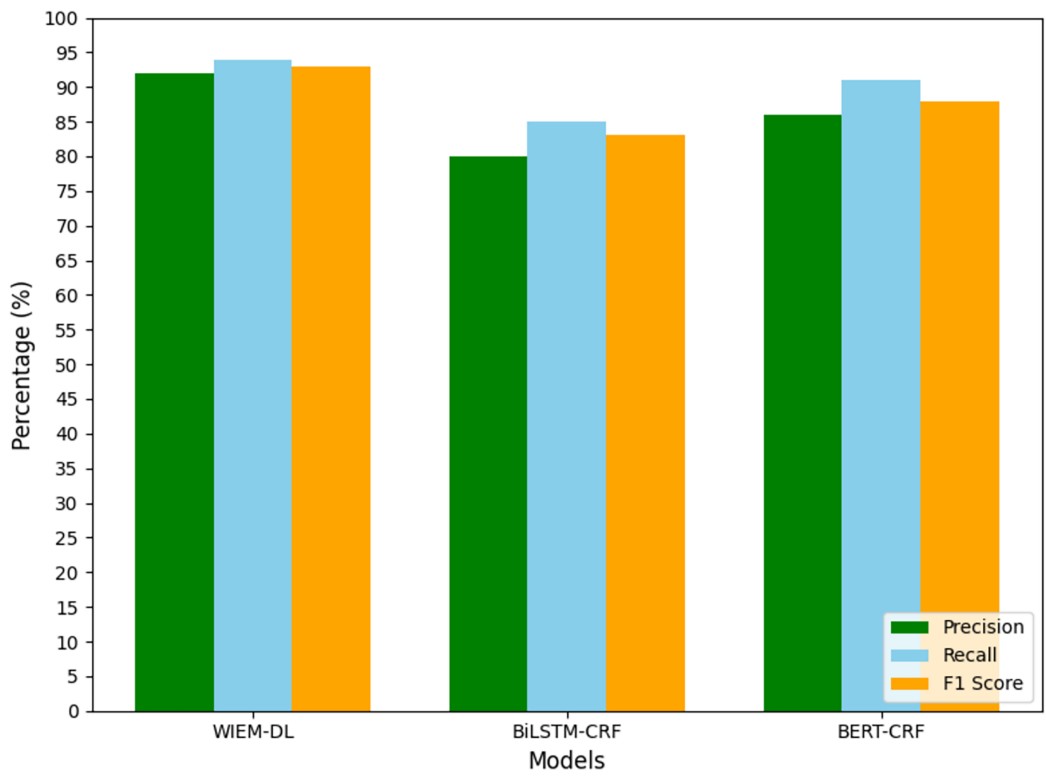

**Figure 8 Sina news information extraction results.**

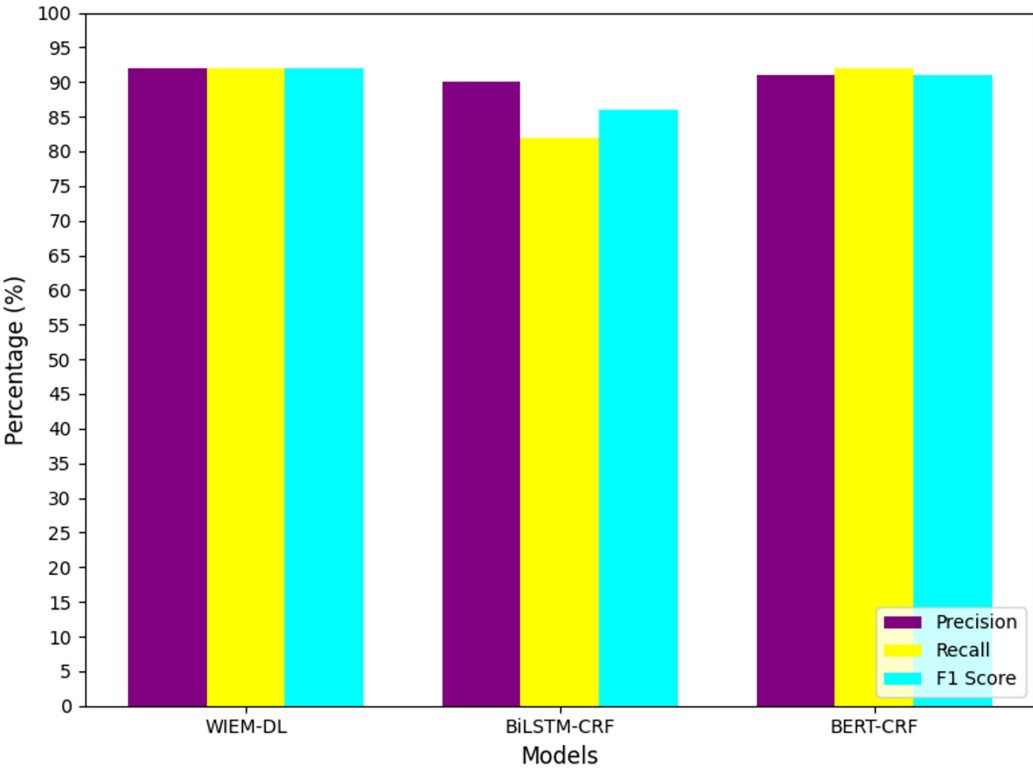

**Figure 9 TouTiao information extraction results.**

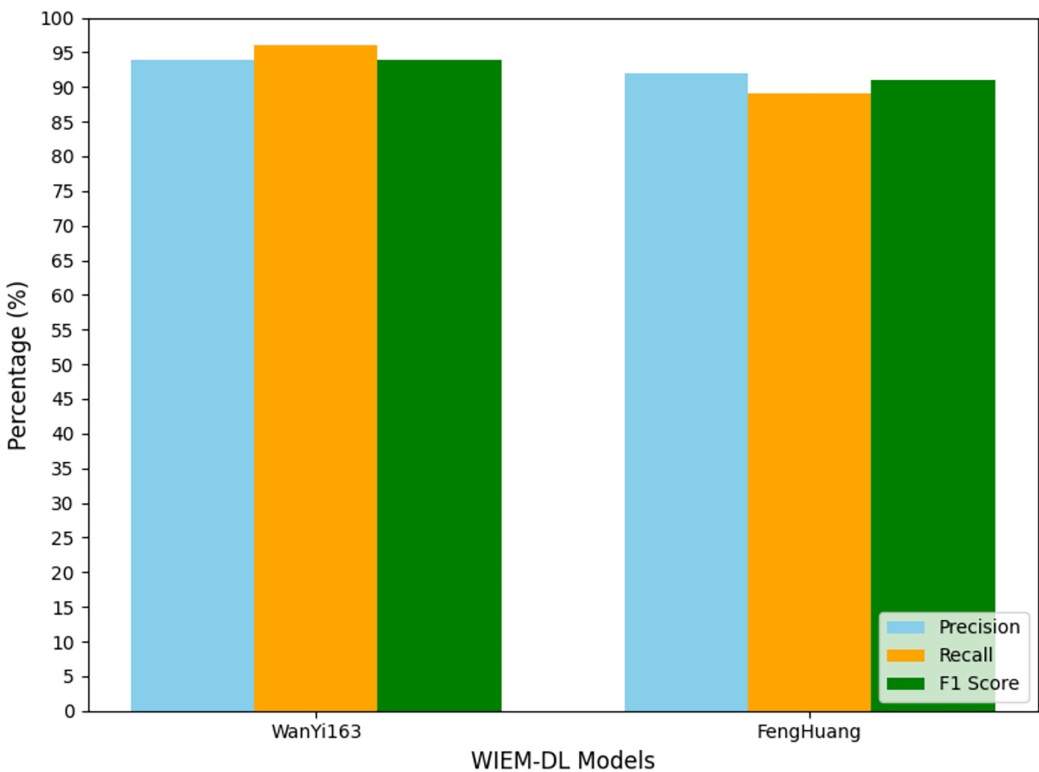

**Figure 10 Transfer training results of information extraction models for two types of websites.**

(3) Ensure comparable training regimes (same hy vperparameters, epochs, and early stopping), Report mean ± CI for each metric.

A large F1 drop when removing sentiment tags indicates high reliance on emotional cues. If the 95% CIs for Full WIEM-DL's F1 do not overlap with those of a variant, the component's effect is statistically significant. Qualitatively examine extraction failures per variant to understand error patterns.

To facilitate data analysis, the statistical results shown in Tables 2–4 were visualized using graphical representations. The final results are illustrated in Figs. 7 to 10. These figures provide a clear comparison of performance metrics, including precision, recall, and F1 score, across different datasets and models, making it easier to evaluate the effectiveness of the WIEM-DL model. Figure 7 presents a comparative analysis of the WIEM-DL model against baseline models such as BERT-CRF and BiLSTM-CRF, focusing on key performance metrics: precision, recall, and F1 score. Figure 8 delves into the performance of the WIEM-DL model across different datasets, such as Sina News, Weibo, and Toutiao. Figure 9 assesses the WIEM-DL model's ability to generalize across different websites, evaluating its performance when trained on one platform and tested on another. Figure 10 presents the results of ablation studies, where specific components of the WIEM-DL

model are systematically removed to assess their individual contributions to overall performance.

### Analysis of cross website information extraction results

In this section, we will analyses the transfer capability of the WIEM-DL model in the task of cross-site web page information extraction through specific experimental design. This experiment uses a small amount of initial labelled data (such as Weibo, Sina News, and Toutiao) to train a cross-site information extraction model that can handle data from different websites. Although these websites contain similar target information (*e.g.*, public opinion topics, sentiment analysis, *etc.*), their web structures differ significantly. Therefore, the model needs to have strong transfer learning ability to apply the knowledge learned from one website to others. Implementing bootstrapping methods to compute 95% CIs for evaluation metrics (precision, recall, F1 score) can provide insights into the reliability of the results. This involves resampling the test data multiple times and calculating the metrics for each sample, thus estimating the range within which the true performance metrics lie with 95% confidence. While WIEM-DL integrates components like BERT embeddings, BiLSTM, and CRF layers, similar architectures have been explored in prior research. Compare WIEM-DL's performance with a wider array of models, including recent state-of-the-art approaches, to contextualize its effectiveness.

## DISCUSSION

In the experiment, the models trained on Weibo, Sina News, and Toutiao were tested through transfer learning to evaluate their performance on the pages from NetEase and Phoenix websites. A set of 1,000 pages from the target websites was used for the experiment to observe the results. The experimental results are shown in Table 4, and Fig. 10 presents the graphical form of the statistical results from Table 4. The WIEM-DL model performed well in cross-website transfer. On the NetEase and Phoenix websites, the F1 scores of WIEM-DL reached 0.95 and 0.94, respectively, indicating that the model can adapt well to different website page structures and effectively extract the required information.

Compared to WIEM-DL, the BERT-CRF and BiLSTM-CRF models performed less effectively in cross-website transfer. On the NetEase and Phoenix websites, their F1 scores were only 0.66 and 0.72, respectively, which were even lower than their performance on the websites they were originally trained on. These results indicate that the knowledge fusion technology and multi-source domain adaptation mechanism of the WIEM-DL model provide it with better adaptability and generalization ability in cross-website transfer. This is highly useful for practical applications where information needs to be extracted from multiple websites. The WIEM-DL model deeply explores and reveals the semantic connotations and relationships of various resource objects in online public opinion big data, constructing a large-scale, knowledge-based fusion network of online public opinion big data. Due to the diversity of formats and organizational structures, the semantic and relational complexity of the data, as well as the complexity of distributed collaborative

operations, the organization of online public opinion big data requires an inclusive, universal, semantically and relationally supportive, decentralized, and low-cost open access mechanism.

## CONCLUSIONS

Web information extraction based on deep learning is a core focus of this study. By training a transferable WIEM-DL model, the limitations of traditional information extraction techniques, such as low efficiency and poor scalability, are effectively addressed. The central concept of the WIEM-DL model lies in leveraging deep learning to model web text. By training a neural network, the model learns the semantic and structural information embedded within the text, enabling the extraction of target information effectively. The WIEM-DL model has demonstrated strong performance in the field of information extraction, particularly excelling in processing unstructured textual data, such as web content. Its primary advantage lies in utilizing the capabilities of deep learning to capture the semantic and structural intricacies of the text, thereby enhancing the accuracy and efficiency of extraction. The WIEM-DL model presented in this study is a transferable deep learning-based Web information extraction framework. It is designed to adaptively learn the structures and extraction patterns of different websites, thereby enhancing the accuracy and reliability of information extraction. By extracting and integrating information from various websites, WIEM-DL facilitates the construction of a unified public opinion knowledge base that encompasses public opinion data across multiple domains, offering extensive application value. Experimental results demonstrate that the WIEM-DL model achieves optimal extraction performance on real-world websites, outperforming other models in both extraction accuracy and time efficiency. These findings hold significant value for better understanding and analyzing public opinion big data, as well as enhancing the efficiency of public opinion forecasting and management.

The WIEM-DL model was trained on a relatively small dataset of 10 manually annotated pages. This limited data may not capture the full variability of public opinion expressions across different platforms, potentially affecting the model's generalizability. The training data was sourced from specific platforms (Weibo, Sina News, and Toutiao), which may introduce platform-specific biases. Consequently, the model's performance on data from other platforms with different linguistic styles or structures remains untested.

Future work will focus on further optimizing the WIEM-DL model by integrating domain-specific knowledge and advanced natural language processing techniques. Expanding its adaptability to a broader range of languages and website structures will also be a key goal. Additionally, exploring its application in real-time web monitoring systems and incorporating user feedback for iterative improvements will help to enhance its robustness and usability in practical scenarios. Exploring the deployment of WIEM-DL in real-time web monitoring systems can assess its practical utility and performance in dynamic environments, facilitating timely public opinion analysis. Implementing mechanisms to incorporate user feedback can enable iterative improvements to the model, ensuring its relevance and effectiveness in evolving information landscapes.

### Funding

This work was funded by the National Natural Science Foundation of China (No. 72161035, 72461033), the National Science Foundation of China Funding Project for Department of Education of Shaanxi Province of China (Grant No. 22JC063), Natural Science and Technology Project Plan in Yulin of China (Grant No. 2024-KJZG-QZZL-004, CXY-2022-94, CXY-2022-93) and the Youth Innovation Team of Shaanxi Universities. The funders had no role in study design, data collection and analysis, decision to publish, or preparation of the manuscript.

### Grant Disclosures

The following grant information was disclosed by the authors:
National Natural Science Foundation of China: 72161035, 72461033.
National Science Foundation of China Funding Project for Department of Education of Shaanxi Province of China: 22JC063.
Natural Science and Technology Project Plan in Yulin of China: 2024-KJZG-QZZL-004, CXY-2022-94 and CXY-2022-93.
Youth Innovation Team of Shaanxi Universities.

### Competing Interests

PengJu Wang is an employee of the Office of the Organizational Committee, Yan'an Municipal Committee. The authors declare no other competing interests.

### Author Contributions

- Yanna Feng conceived and designed the experiments, prepared figures and/or tables, and approved the final draft.
- Feng Zhang conceived and designed the experiments, performed the computation work, prepared figures and/or tables, and approved the final draft.
- Yongheng Zhang analyzed the data, authored or reviewed drafts of the article, and approved the final draft.
- Jiangang Dong performed the experiments, prepared figures and/or tables, and approved the final draft.
- PengJu Wang performed the experiments, authored or reviewed drafts of the article, and approved the final draft.

### Data Availability

The raw data and models are available in the Supplemental Files.

### Supplemental Information

Supplemental information for this article can be found online at http://dx.doi.org/10.7717/peerj-cs.2960#supplemental-information.

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
