# Peer review of "A hybrid extraction model for semantic knowledge discovery of water conservancy big data"

_PeerJ Computer Science, doi:10.7717/peerj-cs.2960_

## Round 0.1 · original submission · Major Revisions

In the opinions of reviewers and mine, this paper should undergo a major revision.

**Language Note:** The review process has identified that the English language must be improved. PeerJ can provide language editing services - please contact us at [email protected] for pricing (be sure to provide your manuscript number and title). Alternatively, you should make your own arrangements to improve the language quality and provide details in your response letter. – PeerJ Staff

Reviewer 1 ·

Basic reporting

This manuscript presents a novel deep learning-based model (WIEM-DL) designed to perform cross-website information extraction in water conservancy public opinion analysis. The authors propose an integrated framework combining BERT embeddings, BiLSTM, attention mechanisms, and CRF for entity recognition and sentiment extraction, demonstrating superior performance over existing models like BERT-CRF and BiLSTM-CRF. The work addresses the need for scalable, transferable web information extraction in domain-specific big data environments. The strengths of the manuscript are a followss:

- The problem is well-motivated and relevant to applied NLP and domain-specific public opinion monitoring.

- The WIEM-DL model is reasonably designed using current deep learning components.

- Empirical results show strong performance and the model's ability to generalize across different website structures.

However, there are some issues also available in the literature:

- The manuscript requires revision for grammar, clarity, and fluency. Sentence phrasing and inconsistent terminologies are present in the manuscript.

- The manuscript includes excessive theoretical background and repetitive explanations in methodology without integrating them into the core contribution.

Experimental design

- The training set appears to be based on small dataset, which is not appropriate given the complexity of the task. More detail is needed on annotation procedures, dataset availability, and validation methods to support the reported performance.

- While the WIEM-DL outperforms baselines, the novelty over existing hybrid models is not well-demonstrated.

Validity of the findings

- A deeper analysis of why this configuration works better, including ablation studies and more extensive cross-domain benchmarks, is required.

- Figures and tables are referenced but not clearly described. It would be better if the authors discussed the figures and the results clearly in the text.

Additional comments

The manuscript needs a significant revision before acceptance! There are many major changes required.

·

Basic reporting

1Poor English grammar and awkward phrasing throughout requires thorough editing
2Introduction fails to clearly establish the research gap and significance
3Literature review lacks logical organization
4Figures have inadequate captions and explanations of symbols/abbreviations
55Methodology presentation is redundant and fragmented
6Missing citations for key statements and inconsistent reference formatting

Experimental design

Training hyperparameters (learning rate, batch size, etc.) are mentioned only briefly at lines 660-667 without justification for these choices.
How the 10 manually annotated pages were selected (lines 642-647).
Discuss statistical significance of performance differences, Include confidence intervals for results.

Validity of the findings

no comment

Additional comments

Lines 39-42: Expand on specific challenges in traditional semantic knowledge extraction techniques with concrete examples.
Lines 79-90: This paragraph repeats information from the introduction. Consider restructuring to avoid redundancy.
Lines 124-132: The contribution statements would be stronger if they were more specific about the technical innovations rather than general advantages.
Section on BERT Embedding (lines 414-427): Provide more technical details on how sentiment analysis is integrated with BERT embeddings.
The experimental results section would benefit from ablation studies to show the contribution of each component of the WIEM-DL model.
Tables 3 and 4: Some values appear inconsistent or missing. Review and ensure all data is accurately presented.

Consider adding a limitations and future work section that honestly addresses current shortcomings and potential improvements.

---

## Round 0.2 · accepted · Accept

In the opinions of original reviewers and mine, this revised paper can be accepted now.

Reviewer 1 ·

Basic reporting

In this version of the manuscript, I don't have any further comments. The authors have solved all my comments.

Experimental design

N/A

Validity of the findings

N/A

Additional comments

N/A

·

Basic reporting

Authors have incorporated all the suggested points. I consider it goog to accept it for publish

Experimental design

Authors have incorporated all the suggested points. I consider it goog to accept it for publish

Validity of the findings

Authors have incorporated all the suggested points. I consider it goog to accept it for publish

Additional comments

Authors have incorporated all the suggested points. I consider it goog to accept it for publish